# Venetoclax Overcomes Sorafenib Resistance in Acute Myeloid Leukemia by Targeting BCL2

**DOI:** 10.3390/biology12101337

**Published:** 2023-10-16

**Authors:** Xi Xu, Weiwei Ma, Guo Qiu, Li Xuan, Chong He, Tian Zhang, Jian Wang, Qifa Liu

**Affiliations:** 1Department of Hematology, Nanfang Hospital, Southern Medical University, Guangzhou 510091, China15521251270@163.com (L.X.);; 2Department of Hematology, The Second Affiliated Hospital, School of Medicine, South China University of Technology, Guangzhou 510006, China; 15720152129@163.com; 3Key Laboratory of Stem Cells and Tissue Engineering, Zhongshan School of Medicine, Sun Yat-sen University, Ministry of Education, Guangzhou 510080, China; 4Children’s Medical Center, Sun Yat-sen Memorial Hospital, Sun Yat-sen University, Guangzhou 510080, China; 5Guangdong Provincial Key Laboratory of Malignant Tumor Epigenetics and Gene Regulation, Sun Yat-sen Memorial Hospital, Sun Yat-sen University, Guangzhou 510080, China

**Keywords:** sorafenib, acute myeloid leukemia, chemoresistance, leukemia, BCL2, venetoclax

## Abstract

**Simple Summary:**

Drug resistance is the main cause of sorafenib treatment failure in clinical acute myeloid leukemia (AML) patients, but the mechanism is currently not fully clear. In this study, we analyzed the genetic characteristics of sorafenib-resistant AML cell subclusters using single-cell and bulk transcription data and found that sorafenib-resistant AML cells can promote *BCL2* transcription by activating STAT3. The BCL2 inhibitor venetoclax can enhance the chemotherapy sensitivity of AML cells to sorafenib.

**Abstract:**

Sorafenib, a kinase inhibitor, has shown promising therapeutic efficacy in a subset of patients with acute myeloid leukemia (AML). However, despite its clinical effectiveness, sorafenib resistance is frequently observed in clinical settings, and the mechanisms underlying this resistance as well as effective strategies to overcome it remain unclear. We examined both single-cell and bulk transcription data in sorafenib-resistant and control AML patients and integrated a sorafenib resistance gene signature to predict the sensitivity of AML cells and the clinical outcomes of AML patients undergoing sorafenib therapy. In addition, our drug sensitivity analysis of scRNA-seq data using deconvolution methods showed that venetoclax was effective in targeting sorafenib-resistant AML cells. Mechanistically, sorafenib was found to activate the JAK-STAT3 pathway and upregulate BCL2 expression in sorafenib-resistant AML cells. This upregulation of BCL2 expression rendered the cells vulnerable to the BCL2 inhibitor venetoclax. In conclusion, we developed a platform to predict sorafenib resistance and clinical outcomes in AML patients after therapy. Our findings suggest that the combination of sorafenib and venetoclax could be an effective therapeutic strategy for AML treatment.

## 1. Introduction

Acute myeloid leukemia (AML) is the most prevalent form of leukemia in adults, comprising 80% of all adult leukemia cases [1]. The incidence of AML escalates with age, rising from 1.3 per million individuals under 65 years old to 12.2 per million in those over 65 years old [2,3]. Leukemia cells often display various mutations, for instance, the FMS-like tyrosine kinase 3 (FLT3)-internal tandem duplication (FLT3/ITD) mutation is observed in 30% AML [4,5]. FLT3/ITD triggers multiple proliferation and survival downstream signaling pathways, including STAT5, RAS, MEK, and PI3K/AKT, in leukemic cells [6,7]. Sorafenib, a multikinase inhibitor, targets overactivated FLT3-ITD mutation to suppress AML cells in clinical treatment [5,8]. 

Although sorafenib demonstrates therapeutic efficacy in AML treatment [9], sorafenib resistance continues to pose a significant clinical challenge, and the underlying mechanisms remain unclear. Sorafenib resistance has been suggested to result from either FLT3 mutations or the alternative activation of downstream pathways involved in cell growth and apoptosis inhibition, such as RAS/RAF/ERK, mTOR, MAPK, and GSK3β/MCL-1 [9,10,11,12,13,14,15]. Hence, developing an effective platform to predict the outcomes of sorafenib treatment and enhance its efficacy is critical for AML management. In this study, we established a platform to predict sorafenib resistance and demonstrated that the BCL2 inhibitor venetoclax could overcome sorafenib resistance in AML cells.

## 2. Materials and Methods

### 2.1. Cell Culture

The THP-1 (TIB-202, ATCC, Manassas, VA, USA), U937 (CRL-1593.2, ATCC), HL-60 (CCL-240, ATCC), MOLM13 (C6600, Beyotime, Shanghai, China), K562 (CCL-243, ATCC), and Jurkat (CRL-2898, ATCC) cell lines were cultivated in RPMI 1640 (11875093, Gibco, Grand Island, NY, USA) supplemented with 10% fetal bovine serum (FBS, SH30084, Hyclone, Logan, UT, USA), 1% penicillin–streptomycin (SV30010, Hyclone), 1% HEPES (H3375-25G, Sigma-Aldrich, St. Louis, MO, USA) and 2 mM glutamine (25-005-CI, Corning, Salt Lake City, UT, USA). All cells were maintained in a humidified incubator at 37 °C with 5% CO_2_.

### 2.2. Flow Cytometry

To perform proliferation analysis, 1 × 10^5^ cells were seeded into a 48-well plate and treated with sorafenib (SF, HY-10201, MCE, Romulus, MI, USA) and/or venetoclax (Vene, HY-15531, MCE) where indicated. Cell growth curves were plotted using flow cytometry to count viable cells, and 7-aminoactinomycin D (7-AAD, 420404, Biolegend, San Diego, CA, USA) was used to exclude dead cells. For apoptosis analysis, 1 × 10^6^ cells were seeded into a 24-well plate and treated with SF and/or Vene where indicated. AnnexinV (640907, Biolegend) and 7-AAD were used to stain the cells. FACS assay was conducted using an analyzer (Attune NxT; Thermo Fisher, Waltham, MA, USA), and the data were analyzed using FlowJo software (FlowJo 10.8.1).

### 2.3. Western Blot

An equal number (1 million) of cells were lysed using RIPA buffer (P0013C, Beyotime) containing 1 mM PMSF (A610425, Sangon Biotech, Shanghai, China). The protein extracts were fractionated using 10% SDS-PAGE and transferred to a PVDF membrane (IPVH00010, Merck Millipore, Burlington, NJ, USA). Following blocking with 5% non-fat milk in Tris-buffered saline containing 0.1% Tween-20 (TBST, pH 7.6) for 1 h at room temperature, the membranes were incubated with primary antibodies for BCL2 (1:1000, PTM-5587, PTM BIO, Chicago, IL, USA), cleaved caspase 3 (1:1000, 9661, CST), p-STAT3 (1:1000, 9145, CST), and β-actin (1:1000, AF0003, Beyotime) overnight at 4 °C, as indicated. The membranes were then incubated with secondary antibodies (1:10,000, rabbit W401B, mouse W402B, Promega, Madison, WI, USA) for 1 h at room temperature. The Western blots were detected either using X-ray film or a digital imaging system (Odyssey Fc). Protein levels were quantified using densitometric intensity.

### 2.4. Single-Cell RNA-seq and Pre-Processing

Fresh primary AML cells were isolated using Ficoll (LTS1077, WEST GENE) gradient centrifugation from leftover BM aspirates of clinical routine sampling. Patient characteristics for primary samples used are provided in Appendix A. AML bone marrow samples underwent single-cell capture using the 10 × Chromium 3’ Library and Gel Bead Kit (10 × Genomics). scRNA-seq libraries were generated as per the manufacturer’s protocol on the Illumina NovaSeq 6000 Sequencing System. To generate feature-barcode matrices, Cell Ranger (version 6.1.2) was employed to align reads with the human genome reference sequence (GRCh38) for the Chromium single-cell data. Processed count matrices for each sample were subjected to quality control where cells containing fewer than 1000 or more than 25,000 unique UMI counts or more than 10% mitochondrial RNA content were excluded. The Seurat R package (version 4.3.0) was used for further processing. The SCTransform command in Seurat was implemented to perform normalization. Principal components analysis (PCA) was used for linear dimensional reduction, while the RunTSNE function in Seurat was used to project the data into a 2D visualization space with *t*-distributed stochastic neighbor embedding (tSNE). Graph-based cell clustering was performed using the FindClusters function in Seurat with the appropriate resolution. To remove the batch effect of each scRNA-seq sample, the Harmony R package (version 0.1.1) was utilized.

### 2.5. Analysis of Sorafenib-Resistant Cells

To forecast the cellular constitution of sorafenib-resistant cells in single-cell data, BayesPrism deconvolution was executed utilizing bulk RNA-seq samples obtained from sorafenib-treated AML cells (GSE104594) [16,17]. The Seurat FindMarkers function was employed to detect marker genes for each cell type. Gene set enrichment analysis (GSEA) was conducted with the ClusterProfler package (version 4.6.0) in R. To gauge pathway activity, gene set variation analysis (GSVA) was executed using GSVA (version 1.46.0). All gene sets were sourced from the Molecular Signatures Database (MSigDB).

### 2.6. Identification and Validation of Sorafenib Resistance Genes

To establish a correlation between patient prognosis and significant genes in sorafenib-resistant clusters, we conducted univariate Cox regression analysis on 553 AML patients from the GEO database (GSE37642). We employed the least absolute shrinkage and selection operator (LASSO) regression analysis to compute the coefficient. Finally, we computed the risk score of individual patients utilizing the following formula: risk score = coefficient1 × expression of gene1 + … + coefficientN × expression of geneN. Based on the median risk scores, patients were categorized into a high-risk (HR) or low-risk (LR) cohort. To assess overall survival (OS) between the HR and LR in each cohort, we employed the Kaplan–Meier analysis. To evaluate the precision of the prognostic model, we utilized ROC curves. We validated our model with 140 samples from the LAML project in the Cancer Genome Atlas (TCGA) database using the same model.

### 2.7. Association with Drug Sensitivity

We obtained ex vivo drug sensitivity area under the curve (AUC) scores for acute myeloid leukemia (AML) from BEAT-AML and Lee et al. [18,19]. Pearson correlation analysis was employed to assess the association between sorafenib-resistant cell abundance and AUC values, which were multiplied by −1. The Enhanced Volcano R package (version 1.16.0) and corrplot R package (version 0.92) were utilized to graphically represent the drug response associations. Furthermore, the BayesPrism deconvolution approach was applied to RNA-seq data obtained from 430 AML patients in the Leucegene cohorts [20]. Principal component analysis (PCA) and Leiden clustering were utilized to map and obtain four hierarchical clusters. To calculate the sorafenib resistance score 7 (SF-res-7), 22 sorafenib resistance genes (named SF-res-22) were used as input features for LASSO regression analysis on the first principal component (PC1), which has high expression in SF-res and low expression in SF-sens.

### 2.8. Cell Line Sensitivity to Sorafenib

RNA-seq data for THP1, MV4-11, HL60, U937, MOLM13, K562, Jurkat cell lines, and primary patient cells were obtained from the GEO database (GSE21758, GSE163466, GSE184891, GSE142662, GSE217585, GSE180229, GSE221851, and GSE202222). The raw count matrices were normalized by applying log2 transformation. The ComBat function in the sva R package (version 3.46.0) was utilized to eliminate batch effects.

### 2.9. Statistical Analysis

The data are presented as means ± standard deviation (s.d.). Student’s *t*-tests were employed for comparison between two groups (* *p* < 0.05, ** *p* < 0.01, and *** *p* < 0.001), while one-way ANOVA followed by Dunnett’s tests were used for multiple comparisons (‡ *p* < 0.05, ‡‡ *p* < 0.01, and ‡‡‡ *p* < 0.001). Statistical significance was established at *p* < 0.05.

## 3. Results

### 3.1. Characterization of Sorafenib-Resistant AML Cells

We conducted single-cell transcriptome analysis of bone marrow cells from eight AML patients, dividing the leukemic cells into fifteen subsets (Figure 1a–c). To investigate the sensitivity of distinct clonal subsets of leukemia cells to SF, we downloaded eight bulk RNA sequencing results of four AML patients before and after sorafenib treatment from the GEO database [15]. We performed Bayesian prism deconvolution on the bulk sequencing results using the fifteen defined leukemia subsets and categorized AML subpopulations into three main groups, namely SF-unrelated (C0, C1, C3, C10, C12, and C13), SF-sens (C2, C4, C7, C8, and C14), and SF-res (C5, C6, C9, and C11), by comparing the cluster abundances before and after treatment (Figure 1d). Our results showed an increase in the SF-res group after relapse, in contrast to the initial diagnosis (Figure 1e). Gene expression analysis demonstrated an increase in MT1G, S100A9, S100A12, and CD36, genes participating in the inflammatory response and fatty acid transport, in the SF-res cell group. In contrast, genes such as HLA-DQB1, ZEP36, MAFB, NFKBIZ, and MS4A7, related to immune response and signal transduction, were upregulated in the SF-sens group (Figure 1f). Furthermore, stemness genes including *SOX4*, *BCL2*, and previously reported LSC markers such as *CD82* and *CD99* were significantly higher in SF-res than in the SF-sens group, indicating that SF-res cells possess stronger stemness (Figure 1g). The gene ontology (GO) and KEGG enrichment analysis revealed that MYC targets, E2F signaling pathway, stemness, G2M checkpoint, DNA repair, and pathways related to stem cell differentiation were upregulated in the SF-res group compared with the SF-sens groups, while inflammatory response, apoptosis, TGF-β, and P53 signaling pathway-related genes were downregulated (Figure 1h,i). In parallel, GSEA revealed that hematopoietic stem cells, MYC targets, G2M checkpoint, DNA repair, and E2F targets were activated in the SF-res group, while apoptosis, inflammatory response, antigen presentation, and hypoxia were silenced (Figure 1j). Together, these findings suggest that genes enriched in the SF-res population function in stemness and cell cycle, while genes enriched in the SF-sens population function in immune response and apoptosis.

### 3.2. The Characteristic Genes of the Sorafenib-Resistant Population Can Predict the Prognosis of Patients

To investigate the correlation between gene sets associated with sorafenib resistance and patient prognosis, we conducted univariate Cox regression analysis on genes that were highly expressed in the SF-res group. Our analysis revealed 45 genes that were significantly linked with prognosis (Figure 2a). Utilizing the LASSO algorithm, we developed a patient prognosis assessment model, where the median of the calculated risk score was set as the threshold for distinguishing between the high-risk (HR) cohort and the low-risk (LR) cohort. We used the 553 patients in the GEO database as the training cohorts (TC). The distribution of risk scores (Figure 2b) and the survival status (Figure 2c) of the HR and LR patients in the TC are displayed. We then reduced the number of significant genes to 22 (Figure 2d) and analyzed the overall survival (OS) of patients in the HR and LR cohorts using Kaplan–Meier (Figure 2e, *p* < 0.001). Our analysis revealed that the OS of patients in the HR cohort was significantly lower than that of patients in the LR cohort. We validated the resistance genes using ROC curves, where the area under the ROC curve (AUC) was positively associated with prognostic accuracy. The AUCs for the 1-year, 3-year, and 5-year patient survival rates were 0.684, 0.721, and 0.722, respectively (Figure 2f). These results suggest that the 22 sorafenib resistance genes, named SF-res-22 thereafter, could effectively predict the prognosis of AML patients.

We further validated our prognostic model using the 140 patients in the TCGA database as the validation cohort (VC). The risk score distribution (Appendix A) and survival status (Appendix A) of the HR and LR patients are demonstrated. Additionally, we reduced the number of genes to 22 (Appendix A) and analyzed the OS of the HR and LR cohorts. Kaplan–Meier analysis revealed that the OS of patients in the HR cohort was significantly lower than that of patients in the LR cohort (Appendix A). The AUCs for the 1-year, 3-year, and 5-year survival rates of patients in the VC were 0.694, 0.696, and 0.678 (Appendix A, *p* = 0.007), respectively. The AUC values of TC and VC were both greater than 0.65, which confirmed that the 22 sorafenib resistance genes were effective in predicting the prognosis of AML patients.

To further scrutinize genes linked with sorafenib resistance, we employed our deconvolution method to deduce the abundance of cell types associated with SF sensitivity from 430 AML patient samples gathered in the Leucegene cohorts [20]. The hierarchy of patients was separated based on two principal components: spanning from SF-sensitive to SF-resistant on PC1, and spanning from SF-sensitive to SF-unrelated on PC2 (Figure 2g left). We postulated that deriving a subscore from the aforementioned 22 sorafenib resistance genes to estimate PC1 could serve as a precise tool to predict the drug sensitivity to sorafenib. Therefore, we retrained these genes on PC1 through LASSO regression and derived a sorafenib resistance score composed of seven genes (sorafenib resistance score 7, SF-res-7) [18]. These seven genes were significantly upregulated in SF-res (Figure 2g right) and mainly related to immune regulation and transcription regulation (Figure 2h). Afterward, we used this streamlined gene set to assess patients’ prognosis. We set the median of the calculated risk score as the threshold for distinguishing the high-risk (HR) cohort from the low-risk (LR) cohort to verify whether the SF-res-7 score could predict patient prognosis. Kaplan–Meier analysis revealed that the OS of HR was significantly lower than that of LR (Figure 2i, *p* = 0.041), confirming that the SF-res-7 score can effectively predict the response of patients to different chemotherapy drugs and thus the patient prognosis.

### 3.3. Sorafenib Resistance Gene Set Predicts Sorafenib Sensitivity of Leukemic Cells

To investigate the correlation between sorafenib resistance genes and the susceptibility of leukemia cells to sorafenib, and to validate the predictive efficacy of SF-res-7 and SF-res-22 prognostic models on cell line sensitivity to sorafenib, we examined the RNA-seq outcomes of various leukemia cells from the GEO database. The prognostic model scored distinct leukemia cell lines, with higher scores reflecting greater resistance to sorafenib. Our findings revealed heterogeneous sorafenib sensitivity across different leukemia cell lines, with the highest resistance scores observed in U937, HL60, Jurkat, and FLT3-ITD mutation patients with sorafenib resistance and the lowest scores in MV4-11, MOLM13, K562, and THP-1 (Figure 3a,b, Appendix A). Additionally, we corroborated the sorafenib resistance score through a cell proliferation assay. Notably, U937 (Figure 3c) and HL60 (Figure 3d) exhibited relatively weak inhibitory effects on proliferation after sorafenib treatment. In contrast, Jurkat, despite receiving a relatively high score, demonstrated a substantial sensitivity to the drug (Figure 3e). Furthermore, the resistance outcomes of K562 (Figure 3f), MOLM13 (Figure 3g), and THP-1 (Figure 3h) were consistent with the prognostic model predictions. Our results confirmed that the sorafenib resistance genes we identified in conjunction with the constructed prognostic model facilitated the prediction of leukemia cell sensitivity to sorafenib.

### 3.4. Analysis of the Drug Sensitivity of Sorafenib-Resistant Cells

No effective treatment schemes have been clinically applied for AML patients who are resistant to sorafenib. Therefore, it is crucial to find novel targeted drugs to overcome sorafenib resistance. The BEAT-AML database provides information on gene expression signatures and the corresponding sensitivity of AML samples to clinically available drugs ex vivo [18]. To analyze the drug sensitivity of sorafenib-resistant cells, we integrated the transcriptional profiles of SF-unrelated, SF-sens, and SF-res populations with the BEAT-AML database to generate a drug sensitivity profile. The three populations displayed significant differences in response to different drugs (Figure 4a,b). The SF-sens cells were more sensitive to staurosporine, while SF-res cells were sensitive to lenalidomide and venetoclax.

To ascertain the correlation between SF-res-7 and the drug sensitivity of AML cells, we analyzed the sensitivity of 33 drugs from BEAT-AML (Figure 4c) and 72 drugs from Lee et al. (Figure 4d) [19]. We found that the cell population with a high expression of SF-res-7 in the BEAT-AML database was more sensitive to ABT-737 and venetoclax and relatively less sensitive to dasatinib and saracatinib (Figure 4e). Cells with a high SF-res-7 score, according to the sensitivity comparison by Lee et al., were more sensitive to navitoclax and azacitidine and relatively less sensitive to dasatinib and rapamycin (Figure 4f). Since venetoclax and navitoclax are both BCL2 inhibitors, Through analysis of the BEAT-AML database and Lee et al.’s database, we predicted that the SF-res population was more sensitive to BCL2 inhibition.

### 3.5. Venetoclax Enhanced the Sorafenib Cytotoxicity to Leukemia Cells

The analysis of the drug sensitivity database revealed that venetoclax effectively targeted the sorafenib-resistant population (Figure 4a). To ascertain the efficacy of venetoclax on sorafenib-resistant cells, we administered leukemia cells with a combined treatment of sorafenib and venetoclax. Sorafenib plus venetoclax, compared with either drug alone, significantly inhibited the proliferation of Jurkat (Figure 5a), MOLM13 (Figure 5b), and K562 (Figure 5c), suggesting that venetoclax improved the sensitivity of leukemia cells to sorafenib. Furthermore, the co-administration of these two drugs resulted in a significant increase in apoptosis of leukemia cell lines, including Jurkat (Figure 5d), MOLM13 (Figure 5e), and K562 (Figure 5f), as evidenced by a substantial rise in the proportion of apoptotic cells and the upregulated apoptotic protein, cleaved caspase 3 (Figure 5g). Our results confirmed that venetoclax was capable of enhancing leukemic apoptosis by upregulating the expression of proapoptotic proteins, thereby augmenting the sensitivity of leukemia cells to sorafenib.

### 3.6. The JAK-STAT3 Pathway Is Activated to Upregulate BCL2 in Sorafenib-Resistant Leukemia Cells

To investigate the underlying mechanism of how venetoclax enhances the sensitivity of leukemia cells to sorafenib, we conducted single-cell sequencing analysis and found that the expression level of *BCL2* was higher in the sorafenib-resistant group (SF-res group) than in the SF-unrelated and SF-sens groups (Figure 6a). The Western blot results further confirmed that BCL2 was significantly upregulated in leukemia cells treated with sorafenib (Figure 6b). To gain insight into the mechanism through which sorafenib upregulates BCL2, we performed a GSEA enrichment analysis and found that the JAK-STAT3 pathway was more active in the SF-res group (Figure 6c). The JAK-STAT3 pathway has been shown to increase the intracellular BCL2 content by promoting *BCL2* transcription [21,22]. The p-STAT3 expression was significantly upregulated after sorafenib treatment using Western blot (Figure 6d), suggesting that the JAK-STAT3 pathway might be involved in the resistance of leukemia cells to sorafenib. To confirm the role of the JAK-STAT3 pathway, we treated leukemia cell lines with sorafenib and Stattic, a STAT3 inhibitor, and detected the BCL2 expression using Western blot. We found that sorafenib upregulated BCL2 expression in leukemia cells, while JAK-STAT3 inhibition decreased the expression of BCL2 induced by sorafenib, suggesting that leukemia cells upregulate the expression of BCL2 through the JAK-STAT3 pathway, leading to resistance to sorafenib (Figure 6e–f).

## 4. Discussion

The acquired mutations on FLT3/ITD and FLT3 tyrosine kinase domain (TKD), which activate the proliferation and survival signal pathway, account for the sorafenib resistance [23,24,25]. Furthermore, sorafenib is applicable for treatment in CML and ALL and represents a distinct target compared with imatinib, a tyrosine kinase inhibitor for the BCR-ABL pathway [26,27,28]. Sorafenib suppresses STAT5 to inhibit MCL1 (MCL1 apoptosis regulator, BCL2 family member), which overcomes imatinib resistance in CML cells [29]. 

By analyzing the drug sensitivity of the sorafenib-resistant cell group, we found that it exhibited greater sensitivity to lenalidomide, VX.745, azacitidine, and venetoclax. Lenalidomide is an immunomodulatory drug used in the treatment of chronic lymphocytic leukemia (CLL) and multiple myeloma [30,31]. Lenalidomide may enhance the function of antileukemia immunity by regulating CD8+ T cells [32]. The combination of sorafenib and azacitidine has demonstrated positive therapeutic effects in patients with recurrent FLT3-ITD mutations after transplantation [33]. VX.745, small molecule inhibitors of MAPK, may increase the sensitivity of leukemia cells to sorafenib by inhibiting the previously reported MAPK pathway related to sorafenib resistance [34,35]. The combination of sorafenib with cytotoxic drugs such as doxorubicin and irinotecan significantly eliminates liver cancer cells in vitro and in vivo [36,37]. The combination of PI3K-delta inhibitor and FLT3 inhibitor has also shown synergistic antitumor activity in AML treatment [38].

Venetoclax has been reported to restore the sensitivity of TKI-resistant leukemia cells by inhibiting the MAPK pathway or downregulating BIM expressions [39,40]. Most importantly, we found that sorafenib can inhibit apoptosis by activating the JAK-STAT3 pathway to promote the expression of BCL2 in leukemia cells. The activation of the STAT3 pathway in sorafenib-resistant cells has also been observed in previous studies of hepatocellular carcinoma, indicating that STAT3 inhibition enhances tumor cell sensitivity to sorafenib [41,42]. Our findings suggest that sorafenib combined with venetoclax could potentially be applied in the clinical treatment of AML patients.

## 5. Conclusions

In summary, we analyzed the gene expression characteristics of sorafenib-resistant AML cells in detail and found that its highly expressed gene set can predict sorafenib sensitivity and prognosis in AML patients. In addition, sorafenib-resistant AML cells activate STAT3, promote *BCL2* transcription, and achieve drug resistance. Targeting the activation of BCL2 in drug-resistant AML cells can effectively enhance the sorafenib sensitivity of AML cells.

## Figures and Tables

**Figure 1 biology-12-01337-f001:**
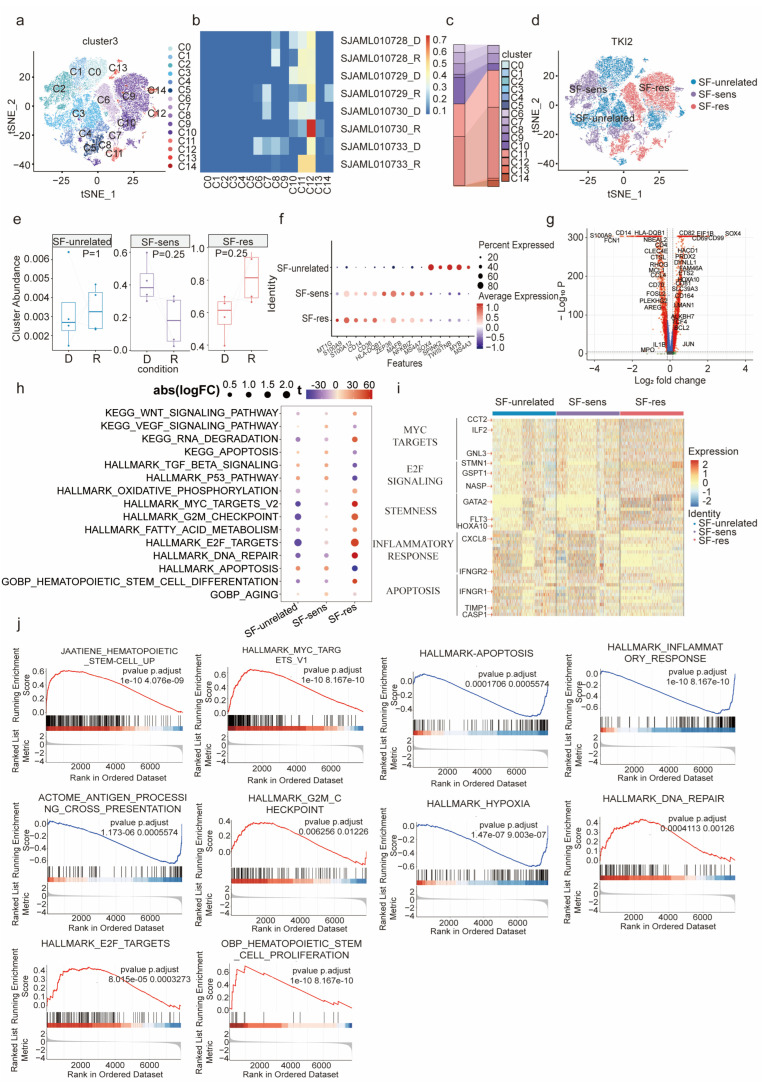
Characterization of sorafenib-resistant AML cells in scRNA-seq: (**a**) The tSNE map was generated by analyzing single-cell RNA sequencing data of bone marrow cells of newly diagnosed (D) and relapsed AML patients (R) were analyzed to generate. (**b**) The distribution of each cluster in 4 pairs of newly diagnosed and relapsed patients. (**c**) Quantification of each cluster of AML patients at diagnosis and relapse. (**d**) The distribution of SF-unrelated, SF-sens, SF-res populations in the tSNE map. (**e**) The abundance of SF-unrelated, SF-sens, and SF-res populations of AML patients at newly diagnosed (D) and relapsed (R). Box plots indicate the range of the central 50% data, with the central line marking the median. Significance was evaluated through a two-sided Wilcoxon rank-sum test. (**f**) Bubble diagram of differentially expressed genes in SF-unrelated, SF-sens, and SF-res clusters, wherein color and size represent the expression level. (**g**) Volcano plot of the differential genes between SF-res and SF-sens cluster. (**h**,**i**) Gene ontology (GO) analysis (**h**) and differentially expressed genes (**i**) in SF-res and SF-sens clusters. (**j**) Gene set enrichment analysis (GSEA) of hematopoietic stem cell up, MYC-target pathway, apoptosis, inflammatory response, antigen-processing cross-presentation, G2M checkpoint, hypoxia, DNA repair, E2F targets, and hematopoietic stem cell proliferation pathways in SF-res and SF-sens clusters.

**Figure 2 biology-12-01337-f002:**
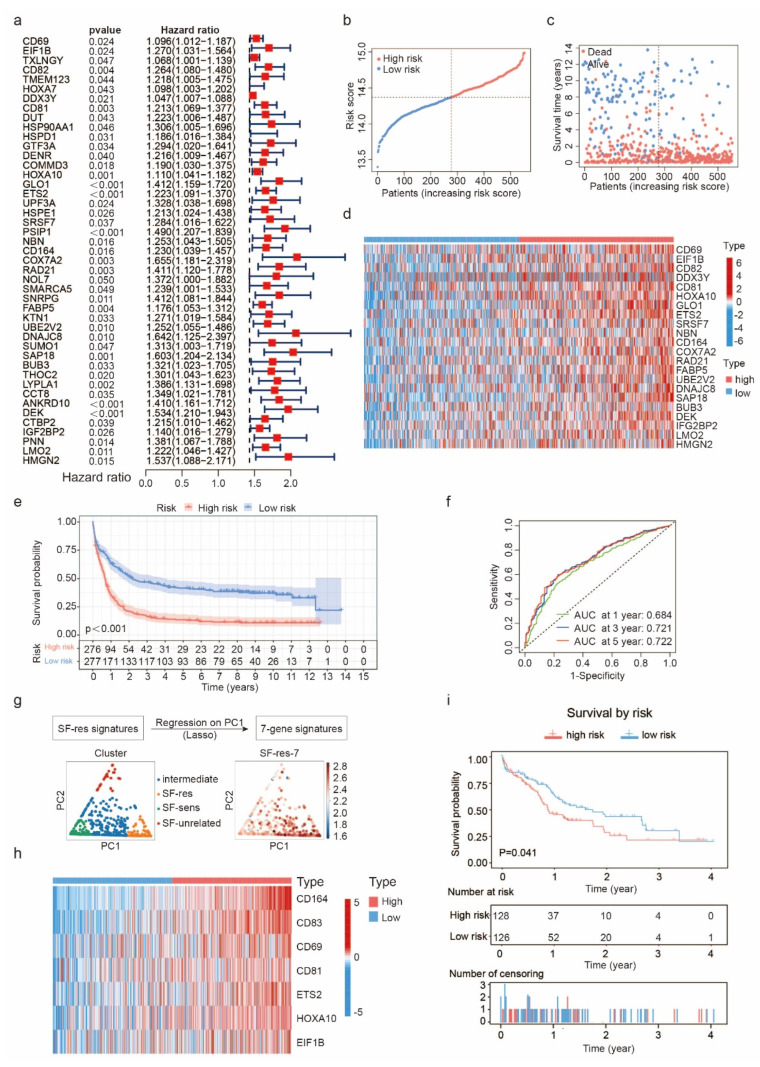
Sorafenib resistance gene set predicts the prognosis of AML patients: (**a**) Forest map shows the relationship between sorafenib resistance gene set and prognosis of patients. (**b**,**c**) The risk score distribution (**b**) and the survival outcome (SO) analysis (**c**) of the training cohorts (TC). (**d**) Heat map of 22 sorafenib resistance genes that are related to AML prognosis in TC. (**e**) The Kaplan–Meier survival curves of the high-risk (HR) and low-risk (LR) patients in the TC. (**f**) The time-dependent ROC analyses of the SF-res-22 prognosis model to estimate the 1-, 3-, and 5-year OS of TC patients. (**g**) PCA of 430 patients with Leucegene cohorts based on the composition of their cellular hierarchy and SF-res-7 (trained on PC1) captures the SF-res versus SF-sens axis. (**h**) Heat map of 7 genes that are significantly upregulated in the SF-res group. (**i**) Event-free survival and relapse-free survival of HR compared with LR patients, stratified by SF-res-7 score into SF-res-7 High (SF-res > SF-sens) and SF-res-7 Low (SF-sens > SF-res). Significance was evaluated through a log-rank test.

**Figure 3 biology-12-01337-f003:**
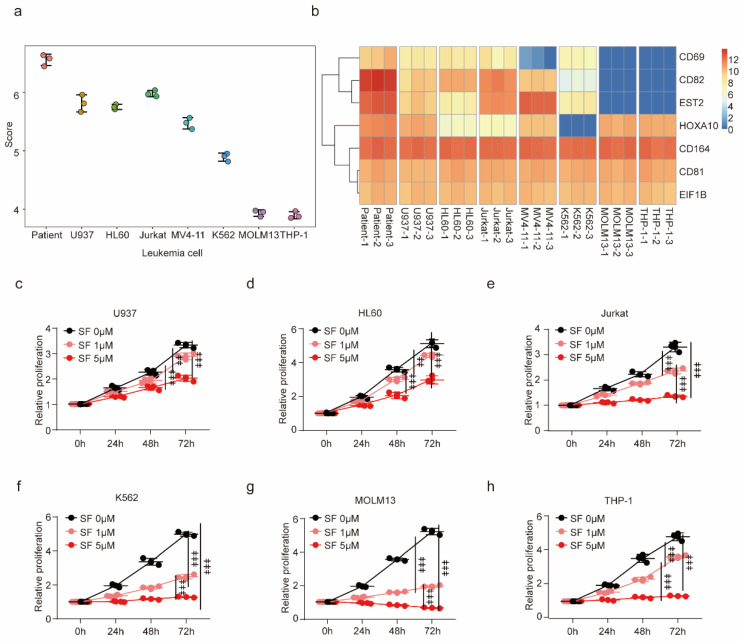
SF-res-7 model predicts sorafenib sensitivity in leukemia cells: (**a**) Analysis of different leukemia cell lines in the GEO database for expressions of the SF-res-7 genes to generate the sorafenib resistance score. (**b**) Heat map of sorafenib resistance genes in different leukemia cells. (**c**–**h**) The growth curve of U937 (**c**), HL60 (**d**), Jurkat (**e**), K562 (**f**), MOLM13 (**g**), THP-1, and (**h**) cells in various dosages of sorafenib (*n* = 3 independent experiments).

**Figure 4 biology-12-01337-f004:**
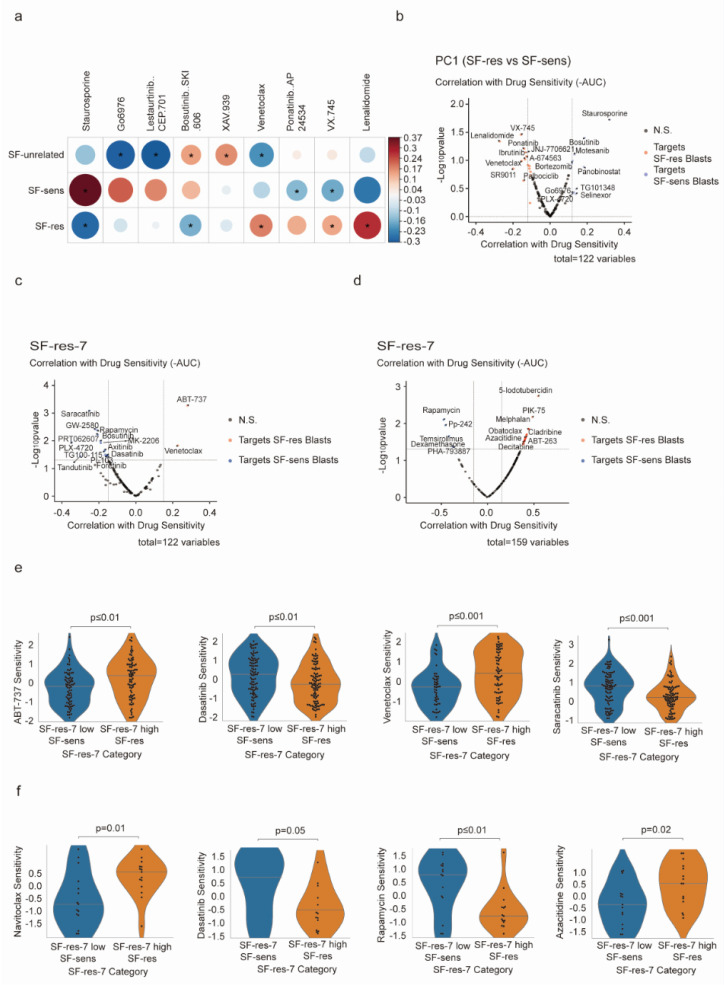
Drug sensitivity screening of sorafenib-resistant cells: (**a**) Pearson correlation showing cell type abundance and ex vivo drug sensitivity (-AUC) across 202 diagnostic patient samples in BEAT-AML, wherein color and size represent the direction and magnitude of the correlation. Only correlations with *p* < 0.05 are presented; those with *q* < 0.05 are marked with an asterisk. (**b**) Volcano plot showing correlations between the SF-res versus SF-sens axis (PC1) and ex vivo drug sensitivities from the BEAT-AML screen, identifying drugs that preferentially target either SF-res or SF-sens AML blasts. (**c**,**d**) Correlation with SF-res-7 identifies drugs targeting either SF-res blasts or SF-sens blasts from BEAT-AML (**c**) (Tyner et al.; *n* =  202) as well as a separate primary AML drug screen (**d**) (Lee et al.; *n* =  30). (**e**,**f**) Violin chart showing drug resistance scores of SF-res and SF-sens populations with the BEAT-AML database (**e**) and a separate prediction model of drug sensitivity (Lee et al.) (**f**).

**Figure 5 biology-12-01337-f005:**
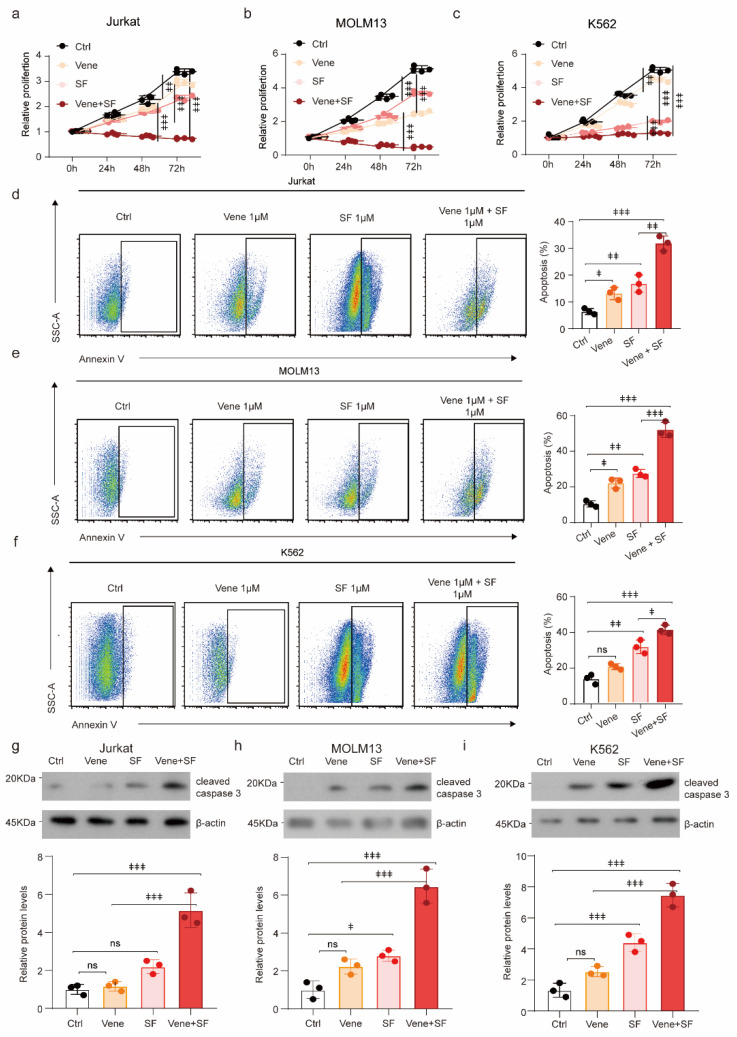
Venetoclax enhances the sensitivity of leukemia cells to sorafenib: (**a**–**c**) The proliferation rate of Jurkat (**a**), MOLM13 (**b**), and K562 cells (**c**) with sorafenib and/or venetoclax treatments as indicated. (**d**–**f**) The representative FACS plots (**left**) and apoptosis rate (**right**) of Jurkat (**d**), MOLM13 (**e**), and K562 cells (**f**) with indicated in vitro treatment for 48 h (*n* = 3 independent experiments). (**g**–**i**) Western blots and quantification of the cleaved caspase 3 in Jurkat (**g**), MOLM13 (**h**), and K562 (**i**) cells received sorafenib (1 μM) and/or venetoclax (1 μM) treatments as indicated. β-actin was used as a loading control. SF, sorafenib; Vene, venetoclax. ns: no significance.

**Figure 6 biology-12-01337-f006:**
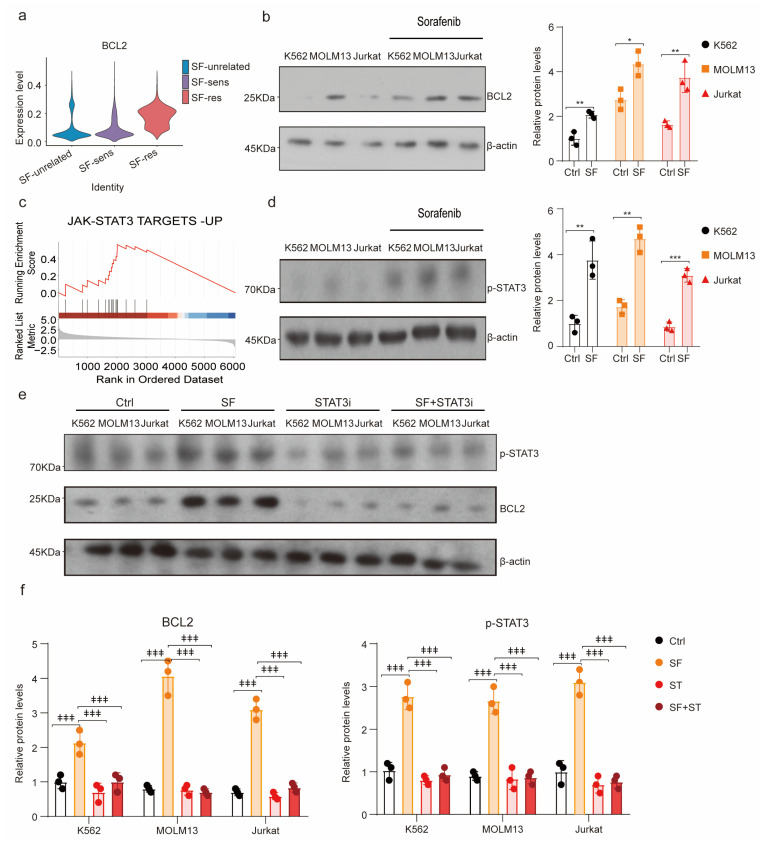
Sorafenib activates the JAK-STAT3 pathway to upregulate BCL2: (**a**) Violin diagram shows the expression levels of BCL2 in SF-unrelated, SF-sens, and SF-res groups. (**b**) Western blots (**left**) and quantification (**right**) of BCL2 in K562, MOLM13, and Jurkat cells with/without sorafenib (1 μM) treatment. β- actin was used as a loading control. (**c**) Gene set enrichment analysis (GSEA) of JAK-STAT3 pathways in sorafenib-resistant group. (**d**) Western blots (**left**) and quantification (**right**) of p-STAT3 in K562, MOLM13, and Jurkat cells with/without sorafenib (1 μM) treatment. (**e**) Schematic diagram of leukemic cells treated with sorafenib and Stattic (10 μM, STAT3 inhibitor). (**f**) Western blots (**f, left**) and quantification (**f, right**) of BCL2 and p-STAT3 in K562, MOLM13, and Jurkat cells with indicated treatment. β-actin was used as a loading control. SF, sorafenib.

## Data Availability

The original contributions presented in the study are included in the article. Further inquiries can be directed to the corresponding authors.

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
