# Peer review of "Venetoclax Overcomes Sorafenib Resistance in Acute Myeloid Leukemia by Targeting BCL2"

_biology, 2023, doi:10.3390/biology12101337_

Round 1
Reviewer 1 Report
The work of Xu et al. shows interesting data for sorafenib resistance in AML primary cells and a panel of cell lines. Sorafenib upregulates JAK-STAT3 signaling in resistant cells responsible for BCL2 expression and cells can be sensitized by venetoclax to cell death.
The study is quite original and shows sensitivity/resistance profiles across FLT3-ITD mutated and wild type cells. Overall the paper is well written and concise and could be of clinical interested for the use of sorafenib in AML treatment.
Issues:
1. BCL2, BCL2 protein and gene nomenclature should be harmonized
2. Overall check typos (ex pp1 ll 18 transcripton(a) and others)
3. In introduction please specify that sorafenib is exclusively active in FLT3-ITD mutated cells but not in FLT3-TKD mutated cell
4. Figure 3 please validate the predictor gene set in at least 1 other FLT3-ITD mutated cell line (MV4-11) and primary patient cells
5. Please provide a table with patient characteristics for primary samples used including the WHO 2022 classification
Author Response
The work of Xu et al. shows interesting data for sorafenib resistance in AML primary cells and a panel of cell lines. Sorafenib upregulates JAK-STAT3 signaling in resistant cells responsible for BCL2 expression and cells can be sensitized by venetoclax to cell death.
Please see the author response in the attached file and revision places are marked with blue in manuscript.

Reviewer 2 Report
Xu and colleagues studied the sorafenib resistance in a model of Acute myeloid leukemia (AML). By means of single-cell and bulk analysis of transcription data from AML patients they compare the differential expression of resistant and sensitive AML cells. Interestingly relapsed AML overexpressed the antiapoptotic protein member BCL2, and the authors consider the opportunity to include Venetoclax a well-known BCL2 inhibitor to overcome the drug resistance of this particular AML phenotype. In general the article is well structured, easily for reading and comprehension, the references are adequate and relevant, the methods employed are appropriated and complementary, the results are clearly presented and support the conclusions.
However, minor observations were found and must be attended prior acceptance.
L15 AML must be defined here and not in L21
L18 transcriptiona > transcription
L19 blc2> BCL2
L19 Venetoclax abbreviation must be included here as it is the first time of appearance
L21 sorafenib/Sorafenib is use indistinctly please homogenate (e.g.,L15, 17, 18)
L37 AML acronym was already stated
L38 [] must be used instead of () for references
Section 2.1 does the media include HEPES? Antibiotics? Antimycotics?
L59 gbico> Gibco
L64 sorafenib of Venetoclax abbreviations must be defined the first time of appearance
L71 “an equal number of cells” cell number must be defined in order to allow replication of the data
L83 Ficoll density or product must be stated
L146 Fig. 1A-C > Fig. 1a-c as the figure shows lowercase letters otherwise change uppercase the letters in the figures. Same for the rest of the figures.
L147 SF acronym was already stated
Figure 1 quality must be improved to clearly appreciate the content
Figure 3 µm must be changed to µM
Figure 3 the figure legend has to include the method employed for the growth curve. The number of experiments and the # of events collected by sample.
Figure 5 Venetoclax concentration must be stated in the figure legend
Figure 5 µm must be changed to µM
Figure 5d-f time of evaluation must be stated in the figure legend
Figure 5 is cells are dying how does the proliferation assays were performed, does the author only gated live cells? This must be clearly explained, as well as the number of events/sample
Figure 6 the working concentrations for each experiment must be stated clearly in figure legend.
L405 reference>references
Author Response
Xu and colleagues studied the sorafenib resistance in a model of Acute myeloid leukemia (AML). By means of single-cell and bulk analysis of transcription data from AML patients they compare the differential expression of resistant and sensitive AML cells. Interestingly relapsed AML overexpressed the antiapoptotic protein member BCL2, and the authors consider the opportunity to include Venetoclax a well-known BCL2 inhibitor to overcome the drug resistance of this particular AML phenotype. In general the article is well structured, easily for reading and comprehension, the references are adequate and relevant, the methods employed are appropriated and complementary, the results are clearly presented and support the conclusions.
However, minor observations were found and must be attended prior acceptance.
Please see author response in the attached file and revision places are marked with blue

Round 2
Reviewer 1 Report
All issues were addressed properly and the paper could be published in its present form.
Author Response
The work of Xu et al. shows interesting data for sorafenib resistance in AML primary cells and a panel of cell lines. Sorafenib upregulates JAK-STAT3 signaling in resistant cells responsible for BCL2 expression and cells can be sensitized by venetoclax to cell death.
